

# Seismic attenuation and dispersion in poroelastic media with fractures of variable aperture distributions

Simón Lissa[1], Nicolás D. Barbosa[2], J. Germán Rubino[3], and Beatriz Quintal[1]

[1]Institute of Earth Sciences, University of Lausanne, Lausanne, Switzerland
[2]Department of Earth Sciences, University of Geneva, Geneva, Switzerland
[3]CONICET, Centro Atómico Bariloche - Comisión Nacional de Energía Atómica, S.C. de Bariloche, Argentina

**Correspondence:** Simón Lissa (simon.lissa@unil.ch)

**Abstract.** Considering poroelastic media containing aligned periodic fractures, we numerically quantify the effects that fractures with variable aperture distributions have on seismic wave attenuation and velocity dispersion due to fluid pressure diffusion (FPD). To achieve this, realistic models of fractures are generated with a stratified percolation algorithm which provides statistical control over geometrical fracture properties such as density and distribution of contact areas. The results are sensitive to both geometrical properties, showing that an increase in the density of contact areas as well as a decrease in their correlation length, reduce the effective seismic attenuation and the corresponding velocity dispersion. Moreover, no FPD effects are observed in addition to the one occurring between the fractures and the background, in the analysed frequency range, by considering realistic fracture models. We demonstrated that if appropriate equivalent physical properties accounting for the effects of contact areas are employed, a simple planar fracture can be used to emulate the seismic response of fractures with realistic aperture distributions. The excellent agreement between their seismic responses is demonstrated for all incidence angles and wave modes.

## 1 Introduction

Fractures in rocks occur in a wide range of scales (from microscale to continental) and their identification and characterisation are important tasks for several areas such as oil and gas exploration and extraction, production of geothermal energy, nuclear waste disposal, and civil engineering works, among others (Schoenberg and Sayers, 1995; Metz et al., 2005; Tester et al., 2007). Given that seismic waves are known to be significantly affected by the presence of fractures (e.g., anisotropy, attenuation, dispersion, scattering), seismic methods are a valuable tool for detecting and characterising fractures. An important cause of seismic attenuation and velocity dispersion occurs when a fluid-saturated heterogenous rock is deformed by a propagating wave. In the case that the rock heterogeneity is a fracture, this is due to the pressure gradient and consequent fluid pressure diffusion (FPD) that are generated as a result of the compressibility contrast between the fracture and the embedding background, which in turns produces energy dissipation (e.g., Pride and Berryman, 2003; Gurevich et al., 2009; Müller et al., 2010). Furthermore, when two or more fluid saturated fractures intersect, a pressure gradient can also arise between the hydraulically connected fractures, thus resulting in additional energy dissipation due to FPD, but affecting higher frequencies than the previously mentioned phenomenon (e.g., Rubino et al., 2013; Quintal et al., 2014, 2016).





Several authors have studied fracture-related FPD effects on seismic attenuation and velocity dispersion (Brajanovski et al., 2005; Rubino et al., 2013; Quintal et al., 2014; Caspari et al., 2016), as well as on the effective anisotropy (Maultzsch et al., 2003; Tillotson et al., 2014; Amalokwu et al., 2015; Rubino et al., 2016, 2017) and scattering (Nakagawa and Schoenberg, 2007; Barbosa et al., 2016), based on experimental or theoretical studies. A common approach to study seismic attenuation

and the associated velocity dispersion in fluid-saturated fractured media, consists of numerically solving Biot's (Biot, 1941, 1962) poroelasticity equations (Quintal et al., 2011, 2014; Rubino et al., 2013). In such studies, fractures are modelled as very compliant, highly porous, and highly permeable heterogeneities embedded in a much stiffer, less porous and less permeable homogenous background (Pride et al., 2004).

Fractures can be conceptualised as two uneven surfaces in contact, which makes fractures to have variable separation between

their boundaries or walls (e.g., Montemagno and Pyrak-Nolte, 1999; Jaeger et al., 2007). In spite of this, fractures are often approximated by simple thin layers of constant thickness. The validity of such representation relies on assumptions such as incident wavelengths larger than the fracture micro-structure, and specific mechanical properties of the background and the material filling the fractures, among others. In this sense, Hudson and Liu (1999) obtained equivalent mechanical properties for a constant thickness layer by representing a fracture as a plane distribution of circular dry or fluid filled cracks. They found

that the fracture geometrical properties that dominate the mechanical behaviour of the fracture are cracks density and length. Still, their model assumes elastic media and crack or contact areas interactions are neglected. Also in an elastic framework, Zhao et al. (2016) have demonstrated the important role played by cracks interaction on the stress field and, consequently, their effects on the overall rock stiffness. Hudson et al. (1996) proposed an analytical solution for obtaining the equivalent elastic properties for a constant thickness layer by considering a fracture as a plane distribution of circular cracks, accounting for

their interaction and considering fluid or visco-elastic material filling the cracks. However, this model is limited to an elastic background and a single fracture. By considering 2D models and following a poroelastic approach based on Biot's (Biot, 1941) equations, Rubino et al. (2014) numerically studied seismic attenuation and dispersion in fluid-saturated fractured media considering several distributions and densities of fracture contact areas. They found that contact areas have a strong effect on the level of seismic wave attenuation and dispersion, and also that their distribution and material properties play an important role

on the effective seismic response. Nevertheless, they performed 2D simulations which cannot account for realistic distributions of fracture aperture and contact areas. To our knowledge, the analysis of the impact of realistic aperture distribution on seismic attenuation and dispersion due to FDP remains largely unexplored.

Nolte and Pyrak-Nolte (1991) presented a stratified percolation algorithm which can be used for generating fractures with realistic aperture distribution. The proposed methodology allows control over the structure of the fractures in terms of density

and distribution of contact areas. Pyrak-Nolte and Morris (2000) studied the effects that such geometrical fracture properties have on the fracture stiffness and on the fluid flow trough the fractures, by considering normal stress changes. Moreover, they established a relation between the hydraulic and mechanical behaviours of fractures. However, FPD effects of fractures with realistic aperture distribution were not considered in their work.

Following the algorithm proposed by Nolte and Pyrak-Nolte (1991) to generate fractures, we numerically quantify the effects

that fractures with variable aperture distributions have on seismic wave attenuation and stiffness modulus dispersion for a





medium containing aligned periodic fractures. We perform a sensitivity analysis in terms of density and distribution of contact areas for compressional waves with normal incidence. Afterwards, we extend the results for all incidence angles and wave modes by analysing the stiffness matrix coefficients. Finally, we show that the numerical results based on intricate fracture geometries can be approximate using a simpler fracture geometry, like a thin layer of constant thickness (White et al., 1975), with appropriate equivalent fracture properties that account for the effects of the realistic fracture structure.

## 2   Numerical upscaling

To study attenuation and modulus dispersion of seismic waves in a fluid-saturated rock with periodic aligned fractures, we model fractures as poroelastic media embedded in a homogenous poroelastic background. The aperture of the modelled fractures can be spatially variable. In the present work, we refer as open regions of the fracture to the zones where the fracture walls are not in contact (non zero aperture) and are filled with a highly permeable and porous material. Contact areas (zero aperture), on the other hand, are represented by a porous material, having the same properties as the background medium as proposed by Rubino et al. (2014). We use similar material properties as those employed by Rubino et al. (2014), but assuming a lower permeability for the background, which could be representative of a sandstone (Table 1) (Bourbie et al., 1987).

Assuming that the prevailing wavelengths are much larger than the fracture aperture and spacing, we can obtain the effective seismic properties by performing oscillatory relaxation tests by solving the quasi-static poroelastic equations given by Biot (1941) with a finite element scheme. The equations are written in the **u**-$p$ formulation, as proposed by Quintal et al. (2011), but in the space-frequency domain following the approach of Rubino et al. (2009). The effective wave moduli are obtained by applying homogeneous harmonic displacements normal to a boundary of the model in the case of P-waves, or a shear displacement in the case of S-waves. The numerical model, which consists of one horizontal fracture embedded in a homogenous background (Fig. 1 (a), left), is the representative elementary volume (REV) of a medium containing aligned periodic fractures. In the right of Fig. 1 (a), we provided a scheme of the described test for the case of normal incidence of P-wave, in which no normal solid displacements are allowed at the lateral boundaries and at the bottom of the model. Additionally, the model is fully saturated with water and the test is performed at undrained conditions, that is, no fluid flow across the model boundaries is permitted.

### 2.1   Equations of poroelasticity

Biot's [1941] equations in the space-frequency domain and in absence of external forces are

$$\nabla \cdot \boldsymbol{\sigma} = 0, \tag{1}$$

$$i\omega \mathbf{w} = -\left(\frac{\kappa}{\eta}\right)\nabla p, \tag{2}$$

where $\boldsymbol{\sigma}$ is the total stress tensor, $\mathbf{w}$ is the relative fluid displacement vector, $p$ is the fluid pressure, $\omega$ is the angular frequency and $i$ is the complex unity. The material properties $\kappa$ and $\eta$ are the permeability and the fluid viscosity, respectively. Equa-




**Table 1.** Material properties

|  | Background and contact areas | Open regions of fractures |
|---|---|---|
| Grain bulk modulus [GPa] | $K_s$=37 | $K_s$=37 |
| Grain density [g/$cm^3$] | $\rho_s$=2.65 | $\rho_s$=2.65 |
| Porosity | $\phi$=0.1 | $\phi$=0.9 |
| Permeability [mD] | $\kappa$=2.37 | $\kappa$=$10^5$ |
| Dry rock bulk modulus [GPa] | $K_m$=26 | $K_m$=0.02 |
| Dry rock shear modulus [GPa] | $\mu_m$=30 | $\mu_m$=0.01 |
| Fluid bulk modulus [GPa] | $K_f$=2.25 | $K_f$=2.25 |
| Fluid density [g/$cm^3$] | $\rho_f$=1.09 | $\rho_f$=1.09 |
| Fluid viscosity [P] | $\eta$=0.01 | $\eta$=0.01 |

tions (1) and (2), the total balance of forces and Darcy's Law, respectively, are also known as consolidation equations. These equations are coupled through the constitutive equations of the porous medium (Biot, 1962)

$$\boldsymbol{\sigma} = 2\mu_m \boldsymbol{\epsilon} + \mathbf{I}\left(\lambda_m e - \alpha p\right), \tag{3}$$

$$p = -M\alpha e - M\nabla \cdot \mathbf{w}, \tag{4}$$

where $\boldsymbol{\epsilon}$ is the strain tensor, $\mathbf{I}$ denotes the identity tensor, and $e$ is the cubical dilatation given by the trace of the strain tensor. The symbols $\lambda_m$ and $\mu_m$ denote Lamé's parameters of the dry frame, and $\alpha$ and $M$ are Biot's poroelastic parameters given by

$$\alpha = 1 - \frac{K_m}{K_s}, \tag{5}$$

$$M = \left(\frac{\alpha - \phi}{K_s} + \frac{\phi}{K_f}\right)^{-1}, \tag{6}$$

where $K_m$, $K_s$ and $K_f$ are the bulk modulus of the dry frame, the solid grains composing the rock frame, and the fluid, respectively, and $\phi$ is the porosity.

Combining Eq. (2) and Eq. (4), the following expression can be obtained

$$\nabla \cdot \left(-\frac{\kappa}{\eta}\nabla p\right) + \alpha i\omega \nabla \cdot \mathbf{u} + \frac{i\omega p}{M} = 0. \tag{7}$$

Finally, Eq. (1) and (7), with the total stress tensor given by Eq. (3), can be used to express the consolidation equations in terms of the unknowns $\mathbf{u}$ and $p$. In 3-D, the solid displacement vector $\mathbf{u}$ has three directional components and the fluid pressure is a scalar. We solve this system of equations in the weak formulation described by Quintal et al. (2011) under the boundary conditions corresponding to the described relaxation tests, with a direct finite element solver of the software Comsol Multiphysics.




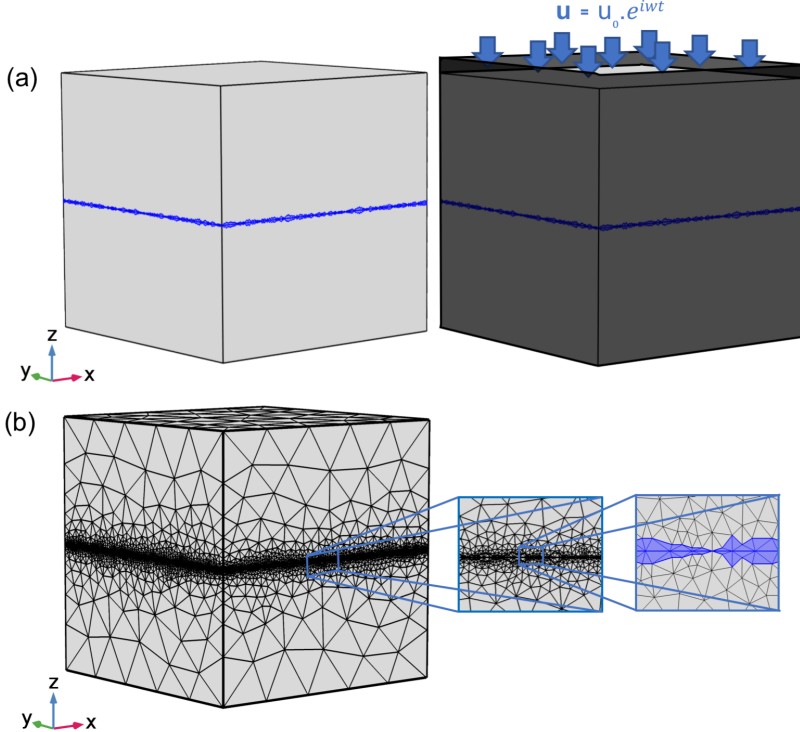

**Figure 1.** (a) REV (left) and oscillatory relaxation test, in which no normal solid displacements are allowed at the laterals and bottom of the numerical model and a vertical displacement is applied at the top of the model (right). (b) Tetrahedral meshing of the numerical model.

## 2.2 Effective properties

The numerical models are discretised in tetrahedral elements (Fig. 1 (b)) where the unknowns **u** and $p$ are calculated, following the methodology described in Section 2.1, for each considered frequency. The stress and strain fields are obtained from **u** and $p$ in each element of the numerical model. For the case of a compressional wave propagating normal to the fractures 5 ($z$-direction), the complex P-wave modulus ($H$) and the P-wave attenuation, estimated as the inverse of the quality factor (O'Connell and Budiansky, 1978), are computed according to the following expressions

$$H(\omega) = \frac{<\sigma_{zz}(\omega)>}{<\epsilon_{zz}(\omega)>}, \qquad (8)$$





$$Q^{-1}(\omega) = \frac{<Im[H(\omega)]>}{<Re[H(\omega)]>}, \tag{9}$$

where $<\sigma_{zz}(\omega)>$ and $<\epsilon_{zz}(\omega)>$ represent the volumetric averages of $\sigma_{zz}$ and $\epsilon_{zz}$ for each frequency and $Re$ and $Im$ corre-

spond to the real and imaginary parts of a complex number (e.g., Masson and Pride (2007); Rubino et al. (2009); Jänicke et al.

5  (2014)).

## 3 Results

In order to numerically analyse the general impact that fractures with variable aperture produce on seismic attenuation and

velocity dispersion, we first consider fractures with simple geometries for representing contact areas and their distributions.

Then, we extend the investigation to fractures of realistic aperture distributions and perform a sensitivity analysis of their

effective seismic response in terms of density and correlation length of contact areas.

### 3.1 Fractures with simple aperture distributions

We first consider simple fracture models for illustrating general effects of contact areas distributions on P-wave modulus

normal to the fractures and the associated seismic attenuation. The numerical model is a cube of 4 cm side having a horizontal

fracture crossing its centre, that is, normal to the vertical (z) direction. This model is the REV of a medium containing a regular

distribution of parallel fractures with almost 4 cm separation between them. Figure 2 shows a representation of the centre plane

of a fracture with regular distribution of contact areas (Fig. 2, left), and another one, with pseudo-random distribution (Fig. 2,

right). The aperture of the open regions of the fracture is constant and equal to 0.4 mm (blue regions), whereas the white square

zones correspond to contact areas (i.e., aperture equal to zero). In both cases, the contact area density is 20% of the fracture

volume.

The numerical results for the real part of the effective P-wave modulus and seismic attenuation normal to the fracture

are presented in Fig. 3. Both models exhibit significant P-wave modulus dispersion and attenuation due to FPD between the

fracture and the background. It means that, when the P-wave compress the fracture, it creates a fluid pressure gradient between

the fracture and the background due to their compressibility contrast. Consequently, a FPD process tends to equilibrate the

pressures producing energy dissipation because of the friction between the fluid and the rock matrix. The responses presented

in Fig. 3 are similar, with the highest dispersion and attenuation observed for the pseudo-random distribution of contact areas.

The largest difference in their P-wave modulus responses is found at the low frequency limit. We compare these results with

the analytical solution of White et al. (1975) for a fracture represented as a thin layer of constant thickness filled with the same

soft material and aperture of 0.4 mm but without contact areas. We observe that the presence of contact areas reduces seismic

attenuation, as they reduce open regions volume thus increasing the fracture stiffness.

For better understanding the impact of contact area distributions on the seismic responses shown in Fig. 3, we plot the

vertical component of the stress field at the low- and high-frequency limits at the centre of the fracture (Fig. 4). The interaction



between contact areas in the pseudo-random case results in a stress shielding and consequently in a decrease in the effective rock stiffness with respect to the regular distribution. At the low-frequency limit, as there is enough time for FPD between the fracture and the background, the effects of interaction between contact areas are maximum because the compressibility contrast between the background and the fracture is maximal. Further, Fig. 3 shows that the real part of the P-wave modulus
for both models converges to a similar value at the high frequency limit. This occurs because there is not time for fluid pressure exchange between the fracture and the background during a half wave period and, therefore, the stiffening effect of the fluid saturating the fracture is maximal. Hence, the effects of contact areas' interaction on the stress field are minimal since the compressibility contrast between the background and the fracture is reduced (Fig. 4).

Summarising, we quantify the relevant impact that contact area distributions have on seismic attenuation and velocity dis-
10 persion due to the interaction between them for simple aperture distributions of fractures.

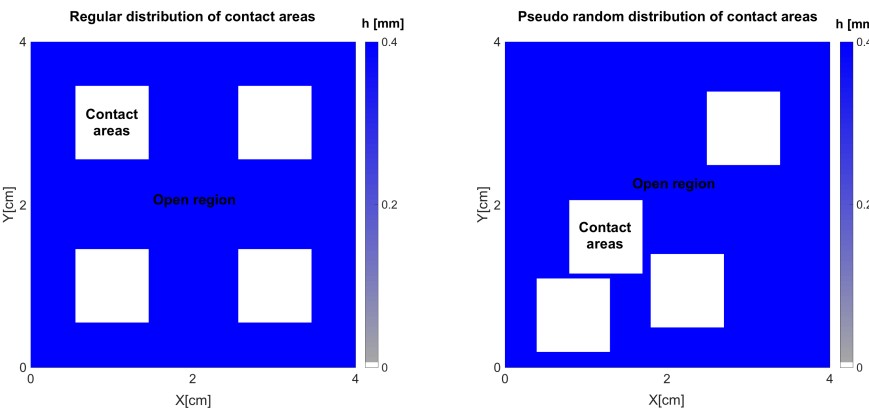

**Figure 2.** Fracture apertures with regular (left) and pseudo-random (right) distributions of contact areas. Blue zones represent open regions of the fracture with 0.4 mm of aperture. Each contact area size is $0.9 \times 0.9$ cm and the contact area density is 20%.

### 3.2 Fractures with realistic aperture distributions

In order to analyse the seismic response of realistic fractures, we performed numerical simulations considering fractures with variable aperture distributions generated following the stratified percolation approach of Nolte and Pyrak-Nolte (1991). Using this approach, a realistic spatial distribution of fracture contact areas can be generated with controlled statistical properties, such
as, correlation length and density (Montemagno and Pyrak-Nolte, 1999). Moreover, it allows us to produce variable aperture distributions of the open regions of the fracture. A description of how the fracture models are generated is given in Appendix A. Using this kind of fracture models, Pyrak-Nolte and Morris (2000) discussed the effects that contact area distributions produce on the mechanical properties of a fracture (i.e., specific stiffness) and they showed that uncorrelated distributions of



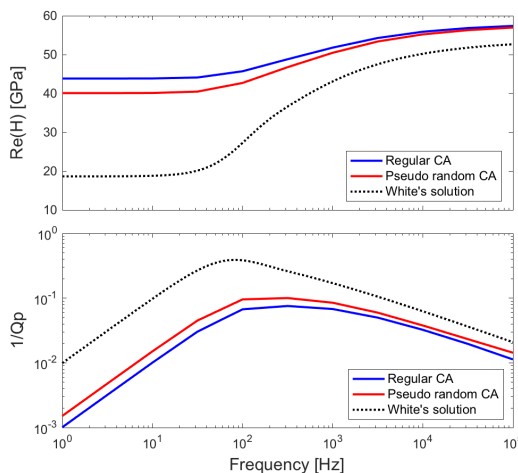

**Figure 3.** Real part of P-wave modulus (H) and attenuation for wave propagation normal to the fracture models illustrated in Fig. 2 as function of frequency.

contact areas produce stiffer fractures than correlated ones. This is in agreement with the results presented in Fig. 3 since a regular distribution of contact areas is analogous to an uncorrelated distribution, considering that the dispersion of distances between contact areas are approximately low for both cases. The aperture of the fractures is given by the vertical distance between the walls which are perfectly symmetrical with respect to the centre of the fracture (Fig. 1). After generation of

5 fractures, the correlation length of contact areas is calculated following the approach of Blair et al. (1993) and it represents an approximation to their average size. Figure 5 shows four fracture aperture distributions chosen for analysing the effects of density and correlation length of contact area on the seismic response. It is important to remark that, if contact area density remains constant, increasing the correlation length of contact areas produces an increase in the mean distance between contact areas, that is, the mean open region length.

In Fig. 6 we observe that a higher contact area density as well as a lower correlation length (model C) produce a stiffer fracture as it can be seen from the relatively high and non dispersive real part of the P-wave modulus. These observations can be partly explained with Fig. 4 as, in this case, a regular distribution of contact areas is equivalent to an uncorrelated distribution. Moreover, these results are in agreement with most analytical solutions, considering that an increase in the correlation length represents also an increase in the mean crack (or open region) length (Hudson and Liu, 1999; Zimmerman and Main,

2004). In such solutions, the excess compliance of a fracture increases with a larger crack length and decreases with a greater contact areas' density. Fracture B has the lowest density and the highest correlation length of contact areas, and as expected, it is effectively the most compliant (Fig. 5). Consequently, it exhibits significant seismic dispersion and attenuation (Fig. 6).




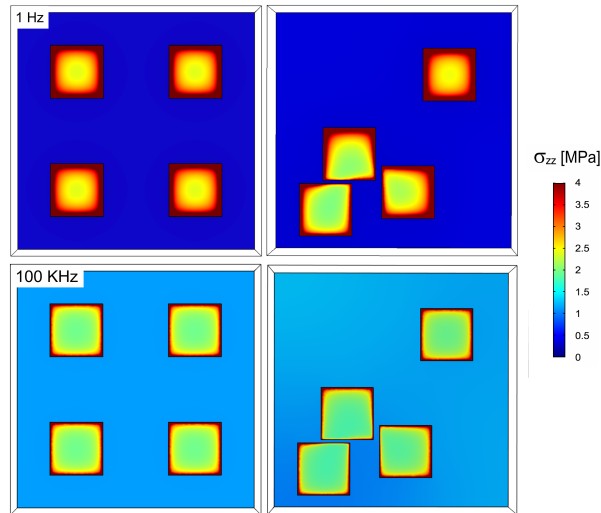

**Figure 4.** Real part of the vertical (z) component of the total stress field at the centre of the fractures illustrated in Fig. 2 at 1 Hz (top) and 100 KHz (bottom).

Interestingly, fractures A and D show comparable seismic responses despite their different geometrical characteristics. This is related to the fact that increasing the correlation length compensates for an increase in contact area density, resulting in fractures with similar effective compliances. Lastly, we note that although the effect of the distribution of contact areas is maximal at the low frequency limit, the distribution also plays an important role in the effective compliance of the rock at the high
frequency limit (Fig. 6).

A common assumption in analytical models is that fracture compliance depends on the fracture volume and on the distribution of the fracture microstructure, such as mean crack radius (e.g., Guo et al., 2017). For analysing the effect of the variable aperture in the open regions of the fractures, we consider simpler fracture models having the same contact areas of those shown in Fig. 5 but setting constant the aperture in the open fracture regions. The aperture is fixed to their mean value (i.e., $0.4$ mm)
(Fig. 7). We refer to this process as binarization of the fractures, since it results in two values for the fracture aperture, zero in the contact areas and $0.4$ mm in the open regions. During this process, the volume of the open regions of the fractures, that is, the volume of the most compliant poroelastic material, remains unchanged. The seismic response are also shown in Fig. 6 and we observe a good agreement between results from fractures with binarized aperture (dashed lines) and those from the previous fractures with more intricate apertures (solid lines), as long as the contact areas remain unchanged. This means that,
the volume of the open regions of the fractures and the distribution of contact areas, are the main characteristics controlling the seismic response, which is in agreement with analytical models.



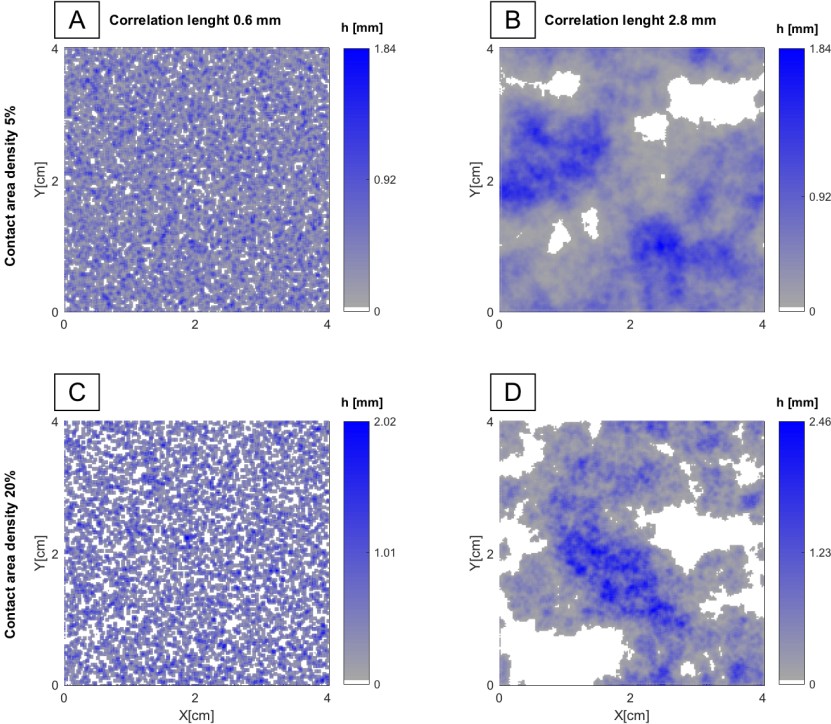

**Figure 5.** Fracture aperture distributions generated using a stratified percolation algorithm (Nolte and Pyrak-Nolte, 1991). Upper models have 5% of contact area density while lower models have 20%. Left models have a small correlation length and right models have a bigger correlation length. All models have a mean aperture of 0.4 mm.

In the analysis presented above, each fracture is one realisation of a pseudo-random generation process with given contact areas density and correlation length. In order to analyse the variability of the results, we generated several fracture models with the same characteristics as model B (Fig. 5), which are referred to as realisations. We choose model B because it exhibits the highest attenuation (Fig. 6) and also, the variability of the results for fractures with uncorrelated distributions of contact areas (such as models A and C) will be lower. For numerical convenience and supported by the comparison shown in Fig. 6, we consider the binarized fracture models of the realisations as in Fig. 7. The fracture models illustrated in Fig. 8 (a) have equal contact area density (5%) and correlation length (2.8 mm) as fracture B. In Fig. 8 (b), the corresponding real part of the P-wave modulus and attenuation, after being binarized, are displayed as black circles for frequencies representative of the relaxed and unrelaxed limits (1 and 100 KHz, respectively) and at the transition frequency (100 Hz). The seismic response

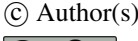



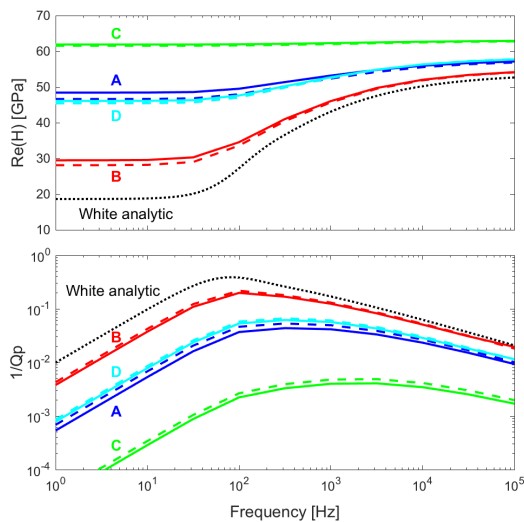

**Figure 6.** Real part of P-wave modulus (H) and attenuation for wave propagation normal to the fractures as function of frequency. Solid lines correspond to fractures with spatially variable aperture in the open regions (Fig. 5) while dashed lines correspond to fractures with binary aperture (Fig. 7).

for model B of Fig. 5 is plotted as a red solid line for reference. The standard deviations of the real part of the P-wave modulus normal to the fractures are presented in Fig. 8 (c) as a function of the number of realisations. We observe larger variability for low frequencies. As we previously illustrated in Fig. 4, this occurs because at the low-frequency limit, pore fluid pressure opposition to compression is minimal and therefore the compressibility contrast between the background and the open

5    regions of the fracture reaches the maximal value. As a consequence, the effects of the interaction between contact areas are more important and then, a change in contact areas geometry, as the one taking place in each realisation, produces a slightly different fracture compliance. As frequency increases, there is less time for fluid pressure communication between fracture and background, the stiffness of the fracture increases and the effects of the interaction between contact areas are less significant, resulting in a smaller variability of the P-wave modulus. For the three analysed frequencies, the variability of the standard

10   deviation became approximately stable after 15 realisations and it is lower than 1 GPa. In other words, it is always lower than 2% of the real-valued P-wave modulus.

### 3.3   Comparison between realistic and simplified equivalent fracture models

In studies on the effective seismic response of fractured media, a fracture is frequently represented as a thin compliant layer of constant thickness (e.g., Schoenberg (1980); Gurevich (2003); Brajanovski et al. (2006); Nakagawa and Schoenberg (2007);



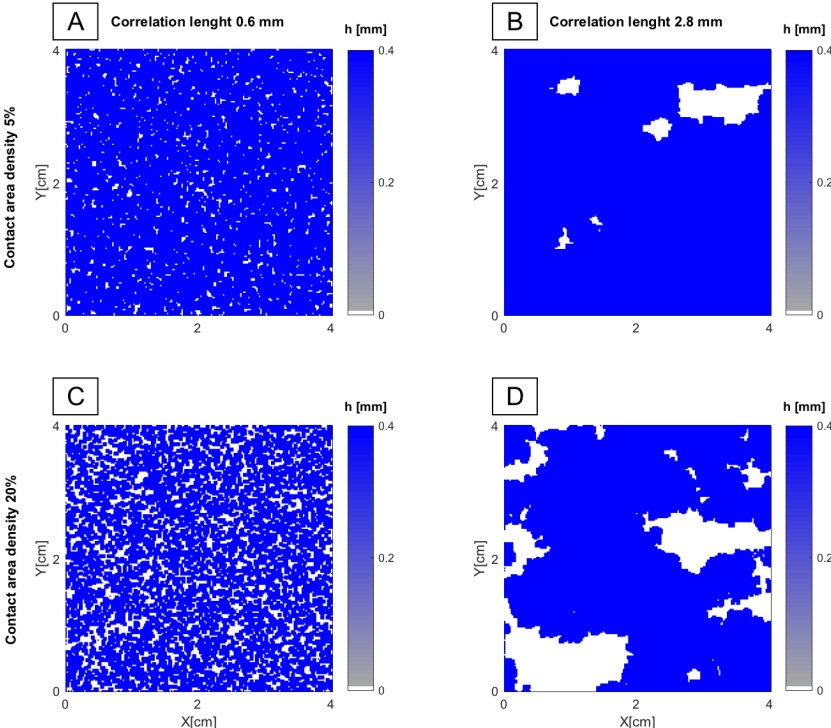

**Figure 7.** Fracture with binary aperture distributions derived from those in Fig. 5. The aperture in the open fracture zone is set equal to the mean value (0.4 mm) of the aperture distribution shown in Fig. 5.

Barbosa et al. (2016)). To analyse the validity of such simple approaches, we estimate equivalent elastic properties of a fracture from the excess compliance computed for realistic fracture models (Fig. 5). Then, we compare the seismic response of such simple fracture model, using the derived equivalent properties, with the numerical results for the fracture models presented in Fig. 5.

5      Following the linear slip theory (Schoenberg and Douma, 1988), the compliance of a dry rock can be split into the contributions from the background and from the presence of fractures, where compliance is defined as the inverse of the stiffness. In this case,

$$\mathbf{S} = \mathbf{S_b} + \mathbf{\Delta S},  \tag{10}$$

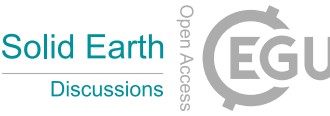



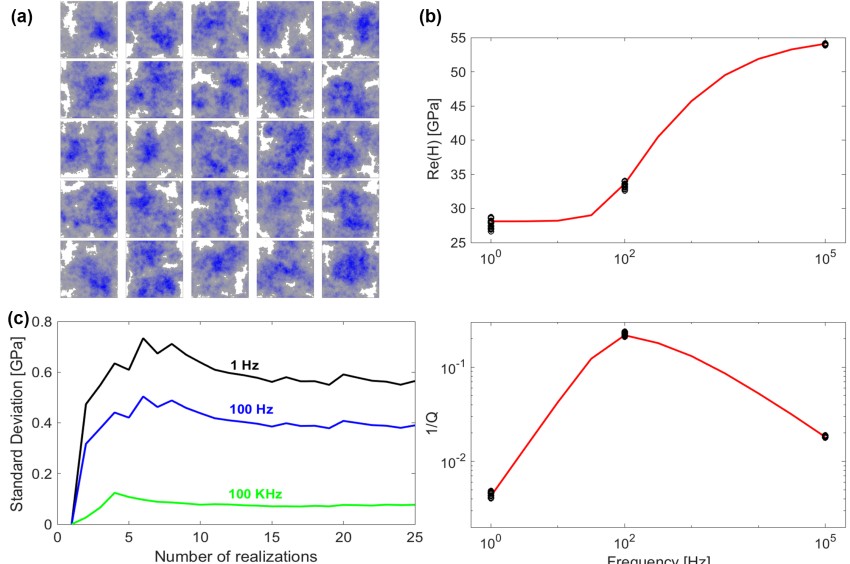

**Figure 8.** (a) Realisations generated with constrains of density and correlation length of contact areas equal to model B. The mean aperture value of all fractures is 0.4 mm. (b) Real part of the P-wave modulus and seismic attenuation normal to the fractures as a function of frequency. The solid red line shows seismic response of fracture B and the black circles correspond to each realisation shown in panel (a) at considered frequencies after being binarized. (c) Standard deviation of the real part of the P-wave modulus as a function of total number of realisations.

where $\mathbf{S}$ is the total compliance matrix with coefficients $S_{ij}$ and $\mathbf{S_b}$ is the compliance matrix of the dry background. The matrix $\mathbf{\Delta S}$ is the excess compliance of the rock due to the presence of fractures which, in turns, is defined as

$$
\mathbf{\Delta S} = \begin{pmatrix} 0 & 0 & 0 & 0 & 0 & 0 \\ 0 & 0 & 0 & 0 & 0 & 0 \\ 0 & 0 & Z_N & 0 & 0 & 0 \\ 0 & 0 & 0 & Z_T & 0 & 0 \\ 0 & 0 & 0 & 0 & Z_T & 0 \\ 0 & 0 & 0 & 0 & 0 & 0 \end{pmatrix}.
\tag{11}
$$



Using Eq. (10) and (11) and the fact that the effective stiffness matrix is the inverse of the compliance matrix, that is $\mathbf{C}=\mathbf{S}^{-1}$, the normal and shear excess compliances $Z_N$ and $Z_T$, respectively, can be obtained as

$$Z_N = \frac{L_m - C_{33}^{dry}}{C_{33}^{dry} L_m}, \tag{12}$$

$$Z_T = \frac{\mu_m - C_{44}^{dry}}{C_{44}^{dry} \mu_m}, \tag{13}$$

where $L_m$ and $\mu_m$ are P-wave and shear moduli of the dry background based on properties given in Table 1. $C_{kk}^{dry}$ (k=3,4) are numerically computed from applying corresponding compressional and shear oscillatory tests to the dry fracture models. Finally, we calculate the equivalent elastic properties from the excess compliances as (Brajanovski et al., 2005)

$$\mu_{fr}^{eqv} = \frac{f_c}{Z_T}, \tag{14}$$

$$K_{fr}^{eqv} = \frac{f_c}{Z_N} - \frac{4}{3}\mu_{fr}^{eqv}, \tag{15}$$

where $\mu_{fr}^{eqv}$ and $K_{fr}^{eqv}$ represent the dry shear and bulk moduli, respectively, of an equivalent fracture of constant thickness and $f_c$ is the fracture volume fraction of the models, given by the ratio between the fracture volume and the REV volume. In this case, as the area in the plane $xy$ of the numerical model is equal to that of the fracture (Fig. 1), then $f_c$ is given by the ratio between the fracture mean aperture and the REV's height (i.e., $f_c = h_{mean}/H$). In order to keep the pore volume of the fractures the same for the equivalent planar fractures, we also calculate the weighted average of the fractures porosities (i.e., fracture equivalent porosity $\phi_{fr}^{eqv}$) for the models shown in Fig. 5 accounting for the effect of contact areas. That is,

$$\phi_{fr}^{eqv} = \phi_b \rho_{ca} + \phi_{fr}(1 - \rho_{ca}), \tag{16}$$

where $\phi_b$ is the background porosity (which is the same as for the contact areas), $\phi_{fr}$ is the porosity of the open regions of the fracture, $\rho_{ca}$ is the contact area density, with $\rho_{ca} = 0.05$ for models A and B and $\rho_{ca} = 0.2$ for models C and D (Fig. 5). Also, $\phi_b = 0.1$ and $\phi_{fr} = 0.9$ as presented in Table 1. Values obtained from Eq. (12)-(16) are shown in Table 2.

In order to support the consideration of our models as representative of realistic fractures, the fracture compliances presented here are qualitatively in agreement with estimated values observed in field and laboratory experiments (Worthington and Lubbe, 2007; Hobday and Worthington, 2012; Barbosa et al., 2018). According to the linear trend proposed by Worthington and Lubbe (2007), our fracture models correspond to lengths ranging from a few meters to tens of meters. Furthermore, it is usually assumed a similar value for the normal and shear compliances (Liu et al., 2000; Sayers, 2002), that is $Z_N/Z_T \approx 1$. Lubbe et al. (2008) used rock samples with artificial fractures to calculate normal and shear compliances from laboratory measurements and they found fracture compliance ratios approaching 0.5. In agreement with their results, the fractures presented in this work exhibit compliance ratios between 0.5 and 1, but closer to 0.5 (Table 2).





**Table 2.** Normal and shear compliances and equivalent properties.

| Model | $Z_N[mPa^{-1}] \times 10^{-12}$ | $Z_T[mPa^{-1}] \times 10^{-12}$ | $K_{fr}^{eqv}[GPa]$ | $\mu_{fr}^{eqv}[GPa]$ | $\phi_{fr}^{eqv}$ | $h_{mean}[mm]$ | $Z_N/Z_T$ |
|-------|------|------|-------|-------|------|-----|-------|
| A | 7.98 | 12.5 | 0.189 | 0.798 | 0.86 | 0.4 | 0.637 |
| B | 39 | 55.2 | 0.015 | 0.181 | 0.86 | 0.4 | 0.705 |
| C | 1.69 | 2.88 | 1.289 | 3.474 | 0.74 | 0.4 | 0.587 |
| D | 9.67 | 13.1 | 0.015 | 0.764 | 0.74 | 0.4 | 0.739 |

### 3.3.1 P-wave modulus analysis

After computing the equivalent dry bulk and shear moduli and porosity calculated for each fracture (Eq. (12)-(16)), as well as the mean aperture, we employ the analytical solution of White et al. (1975) to quantify the P-wave modulus normal to a periodic distribution of constant thickness fractures having these equivalent fracture properties ($\mu_{fr}^{eqv}$, $K_{fr}^{eqv}$, $\phi_{fr}^{eqv}$ and $h_{mean}$).

Figure 9 shows an excellent agreement of the real part of the P-wave modulus and attenuation between fractures with variable aperture distributions (Fig. 5) and equivalent constant thickness fractures. These results show that a very simple fracture geometry, as a thin layer of constant thickness, can approximate much more complicated fracture geometries if appropriate equivalent properties (accounting for contact areas distribution) are used. Most importantly, these results show that there are no additional FPD effects, apart from those occurring between the fracture and the background, produced by the variable aperture

distributions in the considered frequency range.

### 3.3.2 Stiffness matrix generalization

To further verify and generalise the validity of the equivalent fracture model of constant thickness, we extended the methodology presented by Rubino et al. (2016) to numerically compute the effective stiffness matrix from 3D simulations for the realistic fracture models of Fig. 5. To do so, we numerically performed oscillatory relaxation tests following the methodology

described in Section 2, but considering 3 normal relaxation tests (one for each direction $x$, $y$ and $z$) and 3 shear relaxation tests (shearing in the planes $xy$, $xz$ and $yz$). We compare the results with those for fracture of constant thickness with equivalent properties using an analytical solution for a poroelastic medium with transverse isotropy (TI) (Krzikalla and Müller (2011)).

     Figure 10 illustrates the real part of the five independent coefficients $C_{ij}$ for a medium with transverse isotropy (TI) and $C_{66}$, given by $C_{66}$=0.5($C_{11}$-$C_{12}$). The circles in the plots correspond to fractures with variable aperture distribution (Fig. 5),

and solid lines correspond to the analytical solution considering constant thickness fractures with equivalent properties (Table 2). For brevity, fracture C was omitted due to its negligible P-wave modulus dispersion. The stiffness matrix coefficients $C_{11}$ and $C_{33}$ dominate the stress in the medium as a response to a horizontal and vertical compression, respectively ($x$ and $z$ directions). $C_{44}$ and $C_{66}$, on the other hand, dominate the stress as a response to a vertical and a horizontal shear deformation, respectively ($yz$ and $xy$ directions). The coupling coefficients $C_{12}$ and $C_{13}$ are also shown. The effective anisotropy of all the

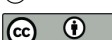



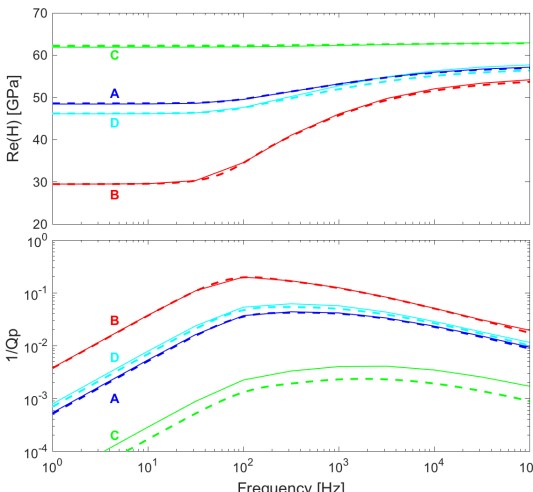

**Figure 9.** Real part of P-wave modulus (H) and attenuation for wave propagation normal to the fractures as functions of frequency. Solid lines correspond to models from Fig. 5 with variable aperture distributions. Dashed lines correspond to fracture models of constant thickness using equivalent fracture properties (Table 2).

models is VTI, although models D and B, due to their high correlation lengths, present a subtle discrepancy between $C_{44}$ and $C_{55}$ below $0.7\ GPa$ (omitted in the figure for brevity). Note that, all the stiffness coefficients of fracture models A and D are similar, which extends the conclusion about the effects of the correlation length and density of the contact areas observed for the normal P-wave modulus to the overall anisotropy of the models (Section 3.2).

5    The excellent agreement between the models with realistic and simple geometries for all the stiffness matrix coefficients generalises our results to all incidence angles and wave modes.

## 4 Discussion

Obtaining information on the hydraulic and mechanical behaviours of fractures by means of their seismic responses is an ultimate goal of fracture characterisation. Pyrak-Nolte and Morris (2000) showed that as response to an increase in normal

10   stress, new contact areas would be created in fractures with a correlated distribution reducing their correlation length and affecting the preferential fluid flow path through the fracture. In this work, we used similar fracture models to study the relation between fracture stiffness and their corresponding seismic response. We showed that the effective seismic attenuation and velocity dispersion due to FPD between the fracture and the background are sensitive to changes in the correlation length and the density of contact areas. This suggests that the effects of FPD on seismic response are potentially affected by normal





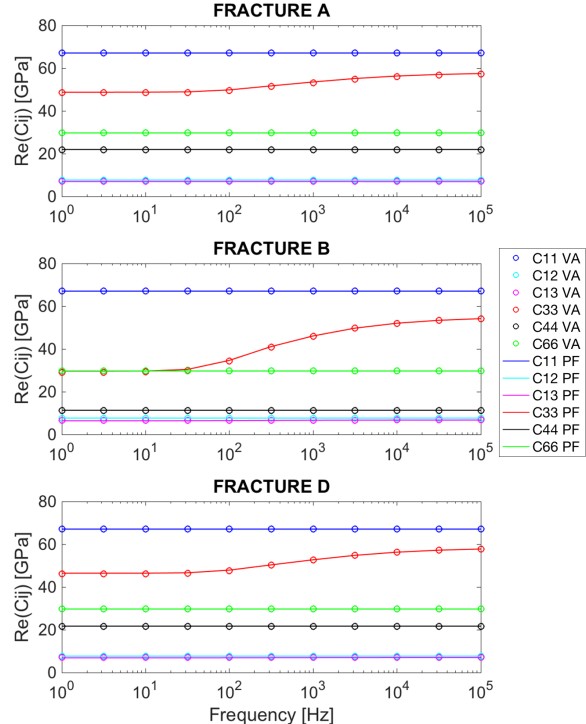

**Figure 10.** Real part of the components of the effective stiffness matrix as functions of frequency. Circles correspond to numerical simulation for fracture models A, B and D with variable aperture (VA) distributions shown in Fig. 5 (fracture model C was omitted due to its negligible P-wave modulus dispersion). Solid lines correspond to the analytical solution of Krzikalla and Müller (2011) for planar fracture (PF) models using equivalent fracture properties (Table 2).

stress variations, and thus, understanding the relation between the fracture stiffness and its corresponding seismic response could represent an opportunity for indirectly monitoring fluid flow changes due to normal stress variations. Rubino et al. (2014) showed that no extra FPD effect due to the presence of contact areas apart from the one between the fracture and the background is observed in the considered frequency range for 2D models. We verified their results for 3D realistic fracture models, and

5 extended the analysis for the effects of spatially variable aperture in the open regions of the fracture for which no extra FPD effects was neither observed.



We also showed that the density and correlation length of contact area control the normal and shear fracture compliances (Fig. 6) and, thus, first order analytical solutions based on these properties, such as the model proposed by Hudson and Liu (1999), could represent a good approximation to the mechanical behaviour of realistic fractures. However, as shown in Fig. 4, the effects of interactions between contact areas must be considered for accurately obtaining the equivalent properties of the material filling the fractures. In this sense, we showed in Fig. 6 and Fig. 10 that a simple layer of constant thickness can successfully reproduce the seismic response of fractures with intricate aperture distributions if equivalent elastic moduli, porosity and aperture are used. However, we remark that such equivalent properties are calculated from the excess compliances of the realistic fractures, and therefore, they account for the significant effects of contact areas distribution and interaction that has been treated in this work. Furthermore, since the linear slip theory is based on the assumption of constant-thickness layer to model fractures (Schoenberg and Douma, 1988; Brajanovski et al., 2005; Guo et al., 2017), our results suggest that this theory can also approximate the seismic response of the realistic fracture models considered in Fig. 5 provided that effective fracture compliances are used.

## 5 Conclusions

The aim of the present contribution was to analyse the effects of variable aperture distributions of 3-D fracture models on FPD between fracture and background. To do so, we numerically quantified the effective frequency-dependent stiffness matrix coefficients and seismic attenuation for realistic fracture models representing REVs of fractured periodic media. Our fracture models were characterised by aperture distributions generated using a stratified percolation algorithm and accounting for different densities and correlation lengths of contact areas. We showed that for a given density of contact areas, fractures with correlated distributions of contact areas (i.e., highest correlation length) exhibit higher P-wave modulus dispersion and seismic attenuation. On the contrary, lower P-wave modulus dispersion and seismic attenuation was observed when increasing contact area density for a given correlation length. This compensatory effect, allows that fractures with highly different geometries produce similar seismic responses. Moreover, although the effects of distribution of contact areas on the P-wave modulus are maximal at the low frequency limit, these distributions also play an important role at the high frequency limit. We also observed that, if the distribution of contact areas is fixed, fracture mean aperture (which controls fracture volume) dominates the seismic response due to FPD effects while the variable aperture in the open regions of the fracture has a negligible influence.

Finally, we demonstrated that a simple fracture geometry such as a thin layer with constant aperture and appropriate equivalent physical properties produces the same effective anisotropic seismic response of a fracture with a much more intricate geometry. The equivalent elastic properties can be obtained from the excess fracture compliances which, in turn, account for the relevant effects of contact areas that we presented here. Our results validate the use of simple constant-thickness fracture models for numerically simulating the effects of fractures with realistic geometries which, in turn, can reduce computational cost and overcome meshing limitations.





## Appendix A: Realistic fracture model generation

To generate the fracture models shown in Fig. 5, we follow the stratified percolation approach described by Nolte and Pyrak-Nolte (1991). To do so, we first define a matrix with zeros in all its cells representing the aperture of a square fracture, whose size is given in terms of the number of cells. The first step consists of stochastically selecting a number ($x$) of the matrix cells. In the

following step, each of those $x$ cells becomes the centre of a new stochastic distribution of $x$ cells confined to a square area around them. The squares are defined by a certain number of cells of the matrix. This process is repeated $n$ times, called tiers, by reducing the size of the squares from one tier to the next one. It means that, if a fix number of cells ($x$) are stochastically located inside each of the squares corresponding to the previous tier, this finally results in selecting $x^n$ cells inside the matrix. In the last tier, which is number $n$ (corresponding to the minimum square size), a value 1 is assigned to each of the $x^n$ selected

cells. Moreover, as there could be overlap between the areas of the squares in all the tiers, and this is a cumulative process, at each time, 1 is assigned to a cell of the matrix and the aperture of the fracture at that cell is increased. In general, a high number of tiers ($n$) with a low number of cells ($x$) creates a correlated distribution of contact areas, which is given by the matrix cells with zero as values. On the other hand, a low number of tiers (n) with a big number of cells ($x$) builds a non-correlated distribution of contact areas (Pyrak-Nolte and Morris, 2000).

The work flow for building up our models consists of importing the matrix generated using this algorithm in Comsol multi-physics, by using an $.stl$ format. The matrix, representing the aperture of the fracture, is converted into a surface and a fracture volume is generated by symmetrically duplicating the fracture surface with respect to their horizontal middle plane. As a result of this process, we obtain a fracture volume with symmetrical walls. Finally, the obtained fracture volume is vertically scaled to yield the desired mean fracture aperture, and embedded in a cubic model (Fig. 1 (a)).

*Competing interests.* The authors declare that they have no competing interests.

*Acknowledgements.* This work has been supported by a grant from the Swiss National Science Foundation. J. Germán Rubino acknowledges a visit to the University of Lausanne financed by the Foundation Herbette.





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
