# Peer review of "Seismic attenuation and dispersion in poroelastic media with fractures of variable aperture distributions"

_Solid Earth, 2019_

## Referee Comment (RC1) · Junxin Guo (Referee) · 15 Apr 2019

This paper investigated the effects of variations of fracture apertures on seismic dispersion and attenuation, which is of great importance for the seismic detection and characterization of fractures. This paper is well written and hence I only suggest minor revisions. The authors may would like to consider the following suggestions:

1. The authors mentioned several times the variable of 'density of contact areas', but no detailed definitions of this variable is given. Since this is one of the major influencing factors that the authors investigated, so the authors should give a clear definition of this variable.

2. Page 4, Line 17: the authors state that 'We solve this system of equations in the weak formulation…'. What does 'in the weak formulation' mean? Please explain this in details.

3. Section 3.1: it seems that the authors choose a REV with only one fracture for the numerical simulations of medium with parallel fractures. This may ignore the boundary condition effects (e.g, Milani et al., 2016, Geophysics) and also the possible fracture interactions. Please comment.

4. Figure 2: the authors consider the contact area to be rectangular, but the contact area in reality can be circular or some other much complicated shapes. What is the possible effects of the contact area shape on seismic attenuation and dispersion? Please comment.

5. Page 15, Line 16: the authors extended the normal incidence case to the oblique incidence case using the approach of Krzikalla and Müller, but no introduction of this approach is given. For the ease of the readers, please give a brief introduction of this approach.

6. Some minor errors that need to be corrected, such as:

   1. Page 2, Line 31: 'have on the fracture stiffness and on the fluid flow trough the fractures', 'trough' should be a typo and should be 'through', please correct.

2. Page 7, Line 5: 'This occurs because there is not time for fluid pressure', 'not' should be corrected to 'no'.

3. Figures 5 and 7: 'Correlation lenght' should be corrected to 'Correlation length'.

4. Figures 6 and 9: Please explain briefly in the figure captions what 'A','B','C', and 'D' in the figures refer to.

Best,

Junxin Guo

---

## Referee Comment (RC2) · Anonymous Referee #2 · 22 May 2019

Review of "Seismic attenuation and dispersion in poroelastic media with fractures of variable aperture distributions" by Simón Lissa, Nicolás D. Barbosa, J. Germán Rubino, and Beatriz Quintal

Dear editor, Dear authors,

The manuscript entitled "Seismic attenuation and dispersion in poroelastic media with fractures of variable aperture distributions " studies the effects of fractures with variable aperture distributions on the seismic attenuation and dispersion.

The authors present several examples were the seismic attenuation and dispersion are computed numerically for material having regular and random aperture distributions.

[Figure]

They discuss the effects on the attenuation of geometrical attributes such as contact area density and correlation length. Following this, the authors discuss the possible usage of analytical models to predict the attenuation and the dispersion in fractured media. They show that simple models can be employed to accurately model materials with relatively complex aperture distributions.

For the most part, the manuscript is Comprehensive and well organized, though confusing at times (see the annotated .pdf). I think it is of interest for the reader of "journal Solid Earth" and would make an interesting contribution.

My main concern is the rather partial and incomplete referencing, there are important references in the field that are missing.

I think the manuscript could be published after carefully addressing the comments in the annotated .pdf.

With Kind regards

Please also note the supplement to this comment:
https://www.solid-earth-discuss.net/se-2019-18/se-2019-18-RC2-supplement.pdf

**Supplement:**

[revised manuscript text omitted]

---

## Author Comment (AC1) · 2 Jul 2019

**Reviewer #1 comments:**

1. (a) *The authors mentioned several times the variable of 'density of contact areas', but no detailed definitions of this variable is given. Since this is one of the major influencing factors that the authors investigated, so the authors should give a clear definition of this variable.*

   (b) Following the reviewer's suggestion, we have added the following definition of "density of contact areas" in Section 2: "We define the density of the contact areas as the ratio between the area of the fracture walls in contact and the area of the entire fracture"

2. (a) *Page 4, Line 17: the authors state that 'We solve this system of equations in the weak formulation…'. What does 'in the weak formulation' mean? Please explain this in details.*

   (b) As indicated in the text, the weak formulation is derived, shown and explained by Quintal et al. (2011). It is out of the scope of our manuscript to go deeply into details about the numerical method.

3. (a) *Section 3.1: it seems that the authors choose a REV with only one fracture for the numerical simulations of medium with parallel fractures. This may ignore the boundary condition effects (e.g, Milani et al., 2016, Geophysics) and also the possible fracture interactions. Please comment.*

   (b) The reviewer points out that the results of the numerical relaxation test applied to the considered REVs may have ignored effects of boundary condition or fracture interactions. The numerical models we employ consist of a cubical background cut by a horizontal fracture which reaches the background boundaries. In the case of planar fractures, such models are essentially unidimensional and as shown by Milani et al. (2016) no boundaries effects would play a role as a consequence of Eq. 1 of our manuscript. However, as the reviewer pointed out, the distribution of the fracture apertures we considered suggests that boundary effects may affect the results. In order to study such possible effects, in Figure I we plot the normalized vertical stress field in the case of the binarized Model B considered in the manuscript (i) for a single repeating unit cell (RUC) and (ii) for 4 RUCs. As shown by Milani et al. (2016), this comparison illustrates boundary condition effects associated with the numerical relaxation test if any. Fig. I shows that the stress fields and consequently, the real parts of the P-wave moduli for both models are not affected by boundary effects. This analysis supports the consideration of our models as REVs of media containing periodically distributed fractures and the fact that no boundary effects or undesired fractures' interaction are affecting the results of the numerical relaxation tests. Moreover, this represents an extension of the results shown by Milani et al. (2016) for fractures that are not unidimensional. We have clarified this in the text in Section 4.

[Figure]

**Figure I**. Real part of the normalized vertical component of the total stress field for wave propagation normal to the binarized model B of the manuscript under dry condition. For a model consisting of 4 RUCs with a zoom to the left bottom RUC (left) and for a simple RUC (right).

**4.** (a) *Figure 2: the authors consider the contact area to be rectangular, but the contact area in reality can be circular or some other much complicated shapes. What is the possible effects of the contact area shape on seismic attenuation and dispersion? Please comment.*

(b) In this example, we only use square contact areas, so that the models are as simple as possible to give the basic understanding of the physical processes. Subsequently, we analyze realistic distribution of fracture apertures in Section 3. We were not interested on studying the effects of the shape of contact areas in very simplified models, but rather to study the effect of contact area distribution in more realistic models.

**5.** (a) *Page 15, Line 16: the authors extended the normal incidence case to the oblique incidence case using the approach of Krzikalla and Müller, but no introduction of this approach is given. For the ease of the readers, please give a brief introduction of this approach.*

(b) The following brief description of the analytical solution from Krizkalla and Müller (2011) has been included in the text: "The analytical solution in based on the relaxed and unrelaxed poroelastic Backus averages of a layered porous medium consisting of a periodic distribution of a stiff background and a soft thin layer. Moreover, they showed that a single relaxation function can be used to link the relaxed and unrelaxed limits of all components

of the stiffness matrix. The corresponding frequency dependence is derived from the P-wave modulus predicted by White et al. (1975). We use such soft layer to approximate a fracture of constant thickness having the equivalent properties ($\mu_{fr}^{eqv}$, $K_{fr}^{eqv}$, $\emptyset_{fr}^{eqv}$ and $h_{fr}^{eqv}$), obtained as described above."

**6.** (a) *Some minor errors that need to be corrected, such as:*
*Page 2, Line 31: 'have on the fracture stiffness and on the fluid flow trough the fractures', 'trough' should be a typo and should be 'through', please correct.*
*Page 7, Line 5: 'This occurs because there is not time for fluid pressure', 'not' should be corrected to 'no'.*
*Figures 5 and 7: 'Correlation lenght' should be corrected to 'Correlation length'.*
*Figures 6 and 9: Please explain briefly in the figure captions what 'A','B','C', and 'D' in the figures refer to.*

(b) All the mentioned minor errors have been corrected.

---

## Author Comment (AC2) · 2 Jul 2019

**Reviewer #2 comments:**

**Page 2.**

**1.** (a) *I think the following reference would be relevant here:*
*Masson, Y. J., and S. R. Pride. "On the correlation between material structure and seismic attenuation anisotropy in porous media" Journal of Geophysical Research: Solid Earth 119.4 (2014): 2848-2870.*

(b) In this part of the manuscript we cite studies on fracture-related fluid pressure diffusion effects on the effective seismic anisotropy of fractured rocks. In the proposed reference, the study shows attenuation anisotropy effects associated with the presence of spheroidal inclusions that could be thought of as idealized fracture or cracks geometries. We have added the corresponding reference as suggested.

**2.** (a) *Please cite the former work:*
*Masson, Yder J., and Steven R. Pride. "Poroelastic finite difference modeling of seismic attenuation and dispersion due to mesoscopic-scale heterogeneity. "Journal of Geophysical Research: Solid Earth 112.B3 (2007).*

(b) In this part of the manuscript we mention studies related with seismic attenuation and velocity dispersion in fluid-saturated fractured media that numerically solve Biot's (Biot, 1941, 1962) poroelasticity equations. The suggested reference has been added here.

**3.** (a) *Numerical simulations based on experimental data have been conducted on fractured rocks, maybe it would be interesting to mention that:*
*Masson, Yder, and Steven R. Pride. "Mapping the mechanical properties of rocks using automated microindentation tests. "Journal of Geophysical Research: Solid Earth 120.10 (2015): 7138-7155.*

(b) In this part of the manuscript we cite studies related with the definition and modeling of fractures. The citation referred by the reviewer presents evidence of the geometry of fractures at the mesoscale on a sandstone sample. Since the suggested reference is relevant for our study, it has been added here.

**Page 3.**

**1.** (a) *I think this reference should be added here:*
*Masson, Yder J., and Steven R. Pride. "Poroelastic finite difference modeling of seismic attenuation and dispersion due to mesoscopic-scale heterogeneity." Journal of Geophysical Research: Solid Earth 112.B3 (2007).*

(b) The suggested reference has been cited in our manuscript (page 6, line 4) but not here as we are specifically referring to oscillatory tests which consider the application of a harmonic displacement as boundary condition.

**2.** (a) *I think Morency and Tromp also used a u-p formulation.*
*Morency, Christina, and Jeroen Tromp. "Spectral-element simulations of wave propagation in porous media." Geophysical Journal International 175.1 (2008): 301-345.*

(b) We have checked the suggested reference and conclude that Morency and Tromp do not employ the *u-p* formulation. Moreover, their study focuses on wave propagation (dynamic equations) while our work considers quasi-static Biot's equations. Thus, we didn't include it in the manuscript.

**Page 6.**

**1.** (a) *Aren't you comparing two different things ? One big fracture vs 4 smaller fractures? Why not use numerical simulations only then ?*

(b) All the curves included in Figure 3 of the manuscript correspond to the effective seismic responses of models containing a periodic distribution of aligned horizontal fractures. Blue and red curves correspond to the models of Figure 2 that have four square contact areas within the fracture and their seismic responses are computed numerically. Dashed lines correspond to the analytical solution of White et al. (1975) for periodically layered media. Since here we aim at illustrating the impact of the presence of contact areas in the fracture, we believe that it is a pertinent comparison and it is representative of effects of including contact areas in a fracture. We modified Figure 2 for increasing clarity by removing the text "open regions" since the descriptive information of the fracture geometries is included in the caption.

**2.** (a) *There is no significant volume effects? i.e. what happens when you double the fracture's aperture ? Maybe it would be interesting to discuss this.*

(b) This is a very interesting comment. We have verified that there are not significant volume effects. In order to demonstrate thus, we have reduced the fracture aperture in the analytical solution (White et al, 1975) by 20%, which makes its fracture volume equal to that of the fractures with contact areas. We included the corresponding curve in Figure 3 of the manuscript. As we can see in the new Fig. 3, such volume reduction only causes a 3.6% and 4% reduction of the P-wave modulus values at the low and high frequency limits, respectively, with respect to the fracture with 20% more volume.

We appreciate the reviewer's comment and, hence, in addition to the above-mentioned change in Figure 3 of the manuscript, we have included the following description: "We compare these results with the analytical solution of White et al. (1975) for a fracture represented as a thin layer of constant thickness filled with the same soft material but without contact areas. Two models are considered, one of a thin layer having constant aperture of 0.4 mm and another one having equal total volume of open regions (that is,

removing the volume occupied by the contact areas) as in the models shown in Fig. 2. We observe that the presence of contact areas reduces seismic attenuation, as they increase the fracture stiffness. For a thin layer with the same volume as the open regions of fractures of Fig. 2 (i.e., a constant thickness of 0.32 mm), we observe minor changes in the seismic response with respect to the thin layer having 0.4 mm of aperture. This means that the contact areas effects on the mechanical behavior of the fracture are more significant than the effects of reducing the open regions volume."

**Page 7.**

**1.** (a) *What is stress shielding? A definition as well as a discussion of its effect would help. Also, changing the geometry (i.e. regular vs random) will change the frequency at which the attenuation peak occurs and how broad it gets. Wouldn't that be sufficient to explain the differences between the two attenuation curves ?*
*Also, it should have an effect on the attenuation levels.*

(b) A definition as well as a short implication of the stress shielding have been added to the text: "The interaction between contact areas in the pseudo-random case results in a stress shielding, which means that the proximity of the contact areas caused a reduction of the stresses on them (Zhao et al., 2016). From Eq. 8, and given that the overall strain in the sample is the same in both cases, it follows that a reduction in the overall stress of the sample translates into a decrease of the effective P-wave modulus in comparison with the regular distribution."

**2.** (a) *It would be interesting for the reader to have a little more information on the stratified percolation model, and, why it is appropriate to model fractures.*

(b) More detailed information about the stratified percolation algorithm, including an explanation of the model generation work-flow can be found in the Appendix A. The use of such algorithm for appropriate modeling of fractures has been shown by Montemagno and Pyrak-Nolte (1999). The reference is included in the text, however this part has been re-phrased: "The consideration of such aperture distribution as realistic is based on the comparison with the imaging of the aperture distribution of a natural fracture network presented by Montemagno and Pyrak-Nolte (1999)"

**Page 8.**

**1.** (a) *I find this very confusing. First I think there is a typo, a regular distribution should be analogous to a correlated distribution. What "dispersion of distances" stands for ? Also approximately low is very vague... This should be rephrased.*

(b) The following detailed explanation about the definition of correlation length and its analogy with the models presented in Figure 2 of the manuscript has been added to text: "After generation of fractures, the correlation length of contact areas is calculated following the approach of Blair et al. (1993), also used by Pyrak-Nolte and Morris (2000), which represents an approximation to their mean length. Consequently, for a given contact area

density, as the correlation length decreases, the fracture exhibits more contact areas with smaller sizes and a narrower distribution of distances between them. Thus, increasing the correlation length of contact areas produces an increase in the mean distance between contact areas, that is, the mean length of the open regions. Using this kind of fracture models, Pyrak-Nolte and Morris (2000) discussed the effects that contact area distributions produce on the mechanical properties of a fracture (i.e., specific stiffness) and they showed that uncorrelated distributions of contact areas produce stiffer fractures than correlated ones. This is in agreement with the results presented in Fig. 3 since a regular distribution of contact areas is analogous to an uncorrelated distribution considering that both have a narrow distribution of distances between contact areas."

Also, some parts of the text have been rephrased for clarifying purpose.

**2.** (a) *Maybe computing the sample's dry stiffness using the same method would worth it to discriminate between poroelastic/elastic effects.*

(b) Model's dry stiffness are an essential component of our study and have been calculated in Section 3.3 to obtain fracture equivalent elastic properties (bulk and shear moduli). Indeed, the differences in effective compliance under dry conditions are given in Table 2 of the manuscript. The relative differences between the considered fracture models have helped us to interpret their poroelastic behavior.

**3.** (a) *Fig 4 isn't an explanation, "can be understood by looking at Fig 4?"*

(b) The review's proposed modification to the text has been made.

**Page 18.**

**1.** (a) *All the examples considered are planar fractures with variable aperture, in this respect the geometry dosen't change much, maybe it would worth it to say a word about general fracture distributions.*

(b) Following the reviewer's comment, "highly different geometries" has been replaced by "highly different aperture distributions".

**2.** (a) *Would that still hold for materials having arbitrary correlation functions ? i.e. frractal materials ?*

(b) On the basis of the excellent agreement between the seismic responses of the linear slip model and the more complex fracture models with variable aperture distributions, we expect that the results we found would be valid also for fractal materials. That is, provided appropriate equivalent fracture properties are used, the aperture variability of the fractures does not produce any additional seismic effects compared with modelling fractures as thin-layers with constant aperture.